# Assessment of Different Circulating Tumor Cell Platforms for Uveal Melanoma: Potential Impact for Future Routine Clinical Practice

**DOI:** 10.3390/ijms241311075

**Published:** 2023-07-04

**Authors:** Arnaud Martel, Baharia Mograbi, Barnabe Romeo, Lauris Gastaud, Salome Lalvee, Katia Zahaf, Julien Fayada, Sacha Nahon-Esteve, Christelle Bonnetaud, Myriam Salah, Virginie Tanga, Stéphanie Baillif, Corine Bertolotto, Sandra Lassalle, Paul Hofman

**Affiliations:** 1Ophthalmology Department, University Hospital of Nice, Cote d’Azur University, 06 000 Nice, France; martel.a@chu-nice.fr (A.M.); nahon-esteve.s@chu-nice.fr (S.N.-E.);; 2Institute for Research on Cancer and Aging, Nice (IRCAN), FHU OncoAge, Cote d’Azur University, 06 000 Nice, France; 3Oncology Department, Antoine Lacassagne Cancer Center, 06 000 Nice, France; 4Laboratory of Clinical and Experimental Pathology, University Hospital of Nice, FHU OncoAge, Cote d’Azur University, Biobank BB-0033-00025, 06 000 Nice, France; 5Inserm, Biology and Pathologies of Melanocytes, Team1, Equipe labellisée Ligue 2020 and Equipe labellisée ARC 2019, Centre Méditerranéen de Médecine Moléculaire, 06 100 Nice, France

**Keywords:** uveal melanoma, liquid biopsy, Cellsearch, ClearCell FX 5, Vortex (VTX-1), ISET, circulating tumor cells

## Abstract

Liquid biopsy and circulating tumor cell (CTC) screening has gained interest over the last two decades for detecting almost all solid malignancies. To date, the major limitation in terms of the applicability of CTC screening in daily clinical practice is the lack of reproducibility due to the high number of platforms available that use various technologies (e.g., label-dependent versus label-free detection). Only a few studies have compared different CTC platforms. The aim of this study was to compare the efficiency of four commercially available CTC platforms (Vortex (VTX-1), ClearCell FX, ISET, and Cellsearch) for the detection and identification of uveal melanoma cells (OMM 2.3 cell line). Tumor cells were seeded in RPMI medium and venous blood from healthy donors, and then processed similarly using these four platforms. Melan-A immunochemistry was performed to identify tumor cells, except when the Cellsearch device was used (automated identification). The mean overall recovery rates (with mean recovered cells) were 39.2% (19.92), 22.2% (11.31), 8.9% (4.85), and 1.1% (0.20) for the ISET, Vortex (VTX-1), ClearCell FX, and CellSearch platforms, respectively. Although paramount, the recovery rate is not sufficient to assess a CTC platform. Other parameters, such as the purpose for using a platform (diagnosis, genetics, drug sensitivity, or patient-derived xenograft models), reproducibility, purity, user-friendliness, cost-effectiveness, and ergonomics, should also be considered before they can be used in daily clinical practice and are discussed in this article.

## 1. Introduction

Uveal melanoma (UM) is the most common primary intraocular malignancy in adulthood [1]. UM remains rarely encountered with an annual incidence of five cases per million inhabitants in Western countries [2]. In the case of localized disease, conservative strategies (e.g., proton radiotherapy, brachytherapy) or primary enucleation are the best treatment [1]. Although primary ocular treatment has been associated with almost 95% of oncologic success [3,4], metastases will occur in about 50% of patients at 10 years [2]. Recent advances in UM pathophysiology have been reported. Of interest, it has been demonstrated that mutations in the *GNAQ*, *GNA11*, and *BAP1* genes act, respectively, as driver and prognostic mutations during UM carcinogenesis [5]. Despite this, there is still no treatment available for managing metastatic spread, and patients usually die within 2 years [6]. UM is genetically different from cutaneous melanoma, explaining why targeted therapies and immunotherapies have been very disappointing in UM until now [5,7]. However, recently, new hopes have been raised by the development of tebentafusp, which is a TCR/Anti-CD3 bispecific fusion protein targeting gp100 [8]. It is noteworthy that tebentafusp can only be administrated to HLA-A*02:01-positive patients, and its overall survival is modest (73% versus 59%) at one year; studies with longer follow-ups (at least 5 years) are warranted and a precise country-by-country medico-economic analysis is still lacking [9].

UM is known to disseminate through the bloodstream and has an exceptional, still unexplained, tropism for the liver [10,11]. Several reliable prognostic factors have been reported, including chromosomal abnormalities (loss of chromosome 3, gain of 8q) [12] and class 2 UMs [13] and have been associated with a metastatic spread and poor overall survival. However, these prognostic factors are usually identified in enucleated specimens or when transvitreal or transscleral tissue biopsy is performed [14]. The use of tissue biopsy in UM is debated because of the risk of extraocular dissemination [14]. Therefore, there is an urgent need to identify reliable and non-invasive biomarkers to identify patients with a high metastatic risk.

Liquid biopsy (LB) is a non-invasive, reliable, and repeatable technique for the diagnosis, prognosis, and follow-up of various solid malignancies [15]. In ocular malignancies, LB allows several tumor-related features to be monitored through the collection of accessible liquids such as blood or aqueous humor [16,17,18]. LB appears particularly promising in UM because (i) the metastatic spread is strictly hematogenous, (ii) tissue biopsy is not routinely performed [14,19], and (iii) detecting circulating tumor cells (CTCs) at the time of diagnosis or during the follow-up may help to predict the risk of subsequent metastatic spread [17]. LB contains many components, including CTCs, circulating free DNA (cfDNA), circulating RNA, small and long non-coding RNAs, ribosomes, exosomes, plasma proteins, and tumor-educated platelets [20,21]. CTCs reflect the tumor’s spatio-temporal heterogeneity through the epithelial-to-mesenchymental transition (EMT), which promotes tumor dissemination, as they remain alive when captured and may be further genetically analyzed and/or cultured [22]. However, compared to cfDNA, CTCs are generally not used in routine clinical practice due to various limitations, in particular some technical issues impairing the sensitivity and specificity of the technique [23,24].

As shown in Table 1, “hunting” a CTC requires three phases which should be individualized: CTC capture, CTC identification, and CTC downstream analyses.

The platforms used for CTC capture are based on label-dependent or label-free properties, explaining the high number of commercially available devices [24]. It should be noted that even when the same CTC platform is used, the pre-analytical phase (blood collection, transport, buffer) as well as the CTC identification method may differ from one study to another one, leading at the end to a lack of consensus on the “ideal” methodology [17,24].

To date, only the immunomagnetic Cellsearch platform (Menarini Silicon Biosystems, Florence, Italy) has been approved—by the Food and Drug Administration (FDA) in 2004—for the diagnosis and prognosis of metastatic breast, colorectal, and prostate cancers [18]. The last decade has been marked by major technological advances, especially in the development of physical microfluidic platforms [24,25]. Ideally, a CTC platform should combine the following features: high sensitivity (high recovery or identification rate), high specificity/purity, high reproducibility, standardized, recovery of viable CTC, user-friendly, rapid processing, ergonomic, and cost-effective. The daily clinical application is often eluded in the literature but is essential for clinicians [17]. To date, only a few studies have compared the efficiency of different CTC platforms [26,27,28,29,30,31,32,33]. To our knowledge, no study has compared different CTC platforms using UM cells [17]. The aim of this study was to compare the efficiency of four different CTC devices by assessing the cell recovery rate for each device using a UM cell line (OMM 2.3). Their potential daily clinical applicability in UM patient management in the near future was then discussed.

## 2. Results

The main results are presented in Table 2 and Figure 1.

The median number of initial cells was, respectively, 42.0 (interquartile range [IQR]: 23.0), 47.0 (IQR: 47.0), 34.0 (IQR: 20.0), and 27.5 (IQR: 10.25) using the Vortex (VTX-1), Cellsearch, ISET, and Cellsearch devices (*p* = 0.014). Pairwise analyses revealed significant differences between the Cellsearch and Vortex (VTX-1) devices (*p* = 0.003) and between the Cellsearch and ClearCell FX devices (*p* = 0.007). No difference was observed between the Vortex (VTX-1) and ISET devices and between the Vortex (VTX-1) and ClearCell FX devices.

The median number of final cells was, respectively, 8.0 (IQR: 12.0), 4.0 (IQR: 5.0), 18.0 (IQR: 17.0), and 0.0 (IQR: 0.0) using the Vortex (VTX-1), Cellsearch, ISET, and Cellsearch devices (*p* < 0.001). Pairwise analyses revealed significant differences between the ClearCell FX and Vortex (VTX-1) devices (*p* = 0.035), between the ISET and ClearCell FX devices (*p* = 0.022), between the Cellsearch and Vortex (VTX-1) devices (*p* < 0.001), between the Cellsearch and ClearCell FX devices (*p* = 0.008), and between the Cellsearch and ISET devices (*p* < 0.001).

The median overall CTC recovery rate was, respectively, 21.2 (IQR: 19.4), 7.7 (IQR: 7.8), 33.3 (IQR: 44.5), and 0.0 (IQR: 0.0) using the Vortex (VTX-1), Cellsearch, ISET, and Cellsearch devices (*p* < 0.001). Pairwise analyses revealed significant differences between the ClearCell FX and Vortex (VTX-1) devices (*p* = 0.019), between the ISET and ClearCell FX devices (*p* = 0.002), between the Cellsearch and Vortex (VTX-1) devices (*p* < 0.001), between the Cellsearch and ClearCell FX devices (*p* = 0.028), and between the Cellsearch and ISET devices (*p* < 0.001). No difference was found between the Vortex (VTX-1) and ISET devices.

Illustrative photographs of the stained CTC collected using the different platforms are presented in Figure 2.

## 3. Discussion

CTC “hunting” has gained interest over the last two decades [34]. New devices allow insight into tumor heterogeneity and downstream single-cell analyses are now possible [22,35,36]. However, the current limitation is the lack of reproducibility regarding CTC capture and identification. This lack of standardization partly explains why CTC screening is not routinely used in daily clinical practice [17,24].

Only a few studies have compared different CTC platforms. This could be explained by the cost of each device and reagent. The Clinical and Experimental Pathology Laboratory (LPCE) located at Nice University Hospital (France) has set up a unique LB platform for CTC detection with four different CTC devices based on immunoaffinity and immunomagnetic (Cellsearch, Menarini, Silicon Biosystems, Florence, Italy) isolation according to the size (ISET, Rarecells, Paris, France) and microfluidic (Vortex [VTX-1], Biosciences, Pleasanton, CA, USA, and ClearCell FX, ClearBridge, Biolidics, Singapore) technologies. For the first time, we compared four different CTC platforms for UM cell isolation and identification.

### 3.1. CTC Recovery Rates

A single UM cell line, cultured in our institution, was used to assess the CTC platforms. Cell lines are often used to compare different methods because they allow a fairer comparison [29,37]. Indeed, cell characteristics, such as size, deformability, and antigen expression levels, were identical when comparing different devices. This allows the experimental conditions to be standardized and the experimental variability to be minimized, especially when comparing the performance of CTC platforms.

We found a significantly higher recovery rate with the ISET (39.2%) and Vortex (VTX-1) (22.2%) platforms compared to the ClearCell FX (8.9%) and Cellsearch (1.1%) devices. Although not significant compared to the Vortex (VTX-1) device, the ISET device was the most sensitive CTC platform in our study.

Our results are in accordance with other studies that have compared various CTC platforms for detecting several solid malignancies [26,28,29,30,31,32,33,38]. As shown in Table 3, the detection rate using the ISET device ranged between 36% and 100%, regardless of the malignancy. Compared to other CTC platforms, the ISET device always provided the highest CTC detection rates (Table 3). With the ISET platform, Mazzini et al. detected CTCs in 55% of cases in 31 primary and metastatic UM patients [39]. Using the same device, Pinzani et al. found a lower detection rate of 31% in 41 UM patients [40], which could be explained by the inclusion of primary UM patients while patients with a metastatic disease were excluded.

The ClearCell FX device is a microfluidic platform for the physical isolation of CTCs based on the inertial focusing process (isolation based on the size, shape, and deformability). In our study, the detection rate obtained with the ClearCell FX device was low (8.9%) and was lower than that obtained with the Vortex (VTX-1) and ISET platforms. Kulasinghe et al. found a detection rate of 47.8% and 51.5% in patients with head and neck cancer and with non-small cell lung cancer, respectively [41]. Yap et al. reported a detection rate of 75.9% in 108 patients with breast cancer [42]. However, these studies were not comparative, and most patients had a metastatic disease. Chudasama et al. found a high detection rate (78.4%) in 32 lung cancer patients [43]. However, the authors only used standard hematoxylin and eosin staining for CTC identification with a high risk of false positives. In their cohort of healthy patients, they found atypical cancer cells in 47%. In our study, the detection rate with the ClearCell FX device was much lower than those reported in the literature. However, comparing our results is challenging because other studies have investigated the efficiency of ClearCell FX using different cell lines and cells from patients with primary and metastatic cancers.

The Vortex (VTX-1) device is a physical microfluidic platform also based on the inertial focusing process [44,45]. In a study published in 2016, the recovery rate of breast carcinoma cell lines was about 38% after two cycles [37]. In patients with advanced prostate cancer, Vortex (VTX-1) identified CTCs in 80% of cases [46]. Our recovery rate (22.2%) was lower than those reported in previous studies. This could be explained by the fact that we were the first to use a UM cell line. Fewer UM cells were used for the experiments (15–150 cells in our experiments vs. ~300 cells usually) despite the fact that we performed two cycles as recommended by the manufacturer [37]. Also, we used a different CTC detection method based on the immunocytochemistry staining of monolayer cultures vs. immunofluorescence [37]. As outlined in Table 3 and in line with our results, Che et al. also found that the Vortex (VTX-1) device allowed higher recovery rates to be achieved (85%) than the Cellsearch device (15%) in 13 patients with breast or lung carcinomas [37].

The Cellsearch platform uses a label-dependent approach for CTC enrichment with automatic capture and identification of CTC. The immunomagnetic-based Cellsearch device is the only FDA-approved CTC platform for metastatic breast, prostate, and colon carcinomas to date [15]. For carcinomas, immunomagnetic antibodies directed against the epithelial cell adhesion molecule (EpCAM), an epithelial cell surface marker, have been used to target CTCs [47]. Melanoma cells do not express EpCAM. The Cellsearch Circulating Melanoma Cell assay uses a human high-molecular-weight-melanoma-associated antigen (HMW-MAA), called Chondroitin Sulfate Proteoglycan 4 (CSPG4 or MCSP), as an identification marker in addition to CD146 (also known as MCAM (Melanoma Cell Adhesion Molecule) or MUC 18) to enrich CTCs from blood [48,49]. Melanoma CTCs must express CD146 and HMW-MAA receptors but not the leukocyte and endothelial cell markers (CD45 and CD34) to be considered CTCs [33]. The advantages of Cellsearch are its high specificity (purity) with positive enrichments allowing leukocyte elimination as well as its full automation, allowing a better reproducibility of the results from one study to another [17,24]. The Cellsearch device has been extensively compared to the ISET device in certain solid tumors and has shown lower recovery rates (Table 3). Several studies have investigated the efficiency of the Cellsearch device in UM [18,48,49,50]. Anand et al. detected CTC in 30% and 42% of primary and metastatic UM patients (n = 39), with a mean number of 5.9 CTCs isolated [49]. In another study conducted in 20 patients, Bande et al. detected CTCs in 50% of primary and metastatic UM patients and none in patients with benign naevi, with a mean number of only one CTC isolated [51]. Terai et al. identified CTCs in 52.9% of 34 primary and metastatic UM patients [50]. Interestingly, they also demonstrated a higher sensitivity in arterial blood samples (detection rate = 100%) compared to venous blood samples (detection rate = 52.9%). Finally, Bidard et al. identified CTCs in 30% of metastatic UM patients (n = 40) and found a positive correlation between the number of CTCs and the presence of miliary liver metastases [48]. In line with Li et al., who identified CTCs in only 1.8% of esophageal cancer patients [38], we found a very low recovery rate (1.1%) using the Cellsearch platform. Of interest, all the experiments were performed externally by the manufacturer itself (Menarini Silicon Biosystems, Bologne, Italy). A positive control (the M229 cutaneous melanoma cell line) was also used to confirm the identification of melanoma cells by the platform. Therefore, we hypothesized that our OMM 2.3 cells did not express CD146 at their surface. CD146 is a cell adhesion glycoprotein strongly expressed by cutaneous melanoma and UM cells. Beutel et al. found that 100% of 35 primary UM cells positively expressed CD146 by immunohistochemistry. However, its expression was highly heterogeneous [52]. They also found a stronger expression of CD146 in metastatic (82.3%) vs. primary (35.4%) UM patients. Lai et al. also demonstrated that CD146 was expressed by seven and three primary and metastatic UM cell lines, respectively, by RT-PCR, immunoblotting, and immunohistochemistry. Immunoblotting has shown variable CD146 levels depending on the cell lines [53]. While the OMM 2.3 cell line has not been analyzed in this study, lower CD146 levels have been found in UM cell lines previously treated with iodine radioactive plaques, such as the OMM 2.3 cell line, compared to untreated cell lines [53]. Similarly, Beasley et al. [54] performed several immunohistochemistry analyses in five different UM cell lines (OMM 2.3 cells were not studied). CD146 expression was highly variable from one cell line to another (from no to moderate marker intensity). ABCB5, MART 1, and GP 100 were the most strongly expressed markers in almost all the studies of UM cell lines. Flow cytometry revealed strong CD146 expression in all investigated UM cell lines. Interestingly, CD146 was not expressed in 6 out of 10 UM patients and MCSP was not detected in any of the UM specimens [54]. Finally, CD146 was expressed in all the 80 UM samples available in The Cancer Genome Atlas analyses [55]. We analyzed CD146 by immunochemistry in OMM 2.3 and M229 cell lines (see Appendix A). The expression was negative in OMM 2.3 cells and positive in M229 cell lines.

To our knowledge, our study was the first to isolate in vitro UM cells using the Cellsearch device. According to us, the variable and heterogeneous expression of CD146 could explain the low recovery rate and the discrepancies found in the literature. Furthermore, a highly variable expression of the antigen in identical cell lines cultured in different institutions has already been reported, resulting in recovery rates ranging between 12 and up to 83% for a given epithelial cell line when the CellSearch system was used [56]. This finding highlights a limitation of the label-dependent Cellsearch platform. Unlike other CTC platforms based on label-free approaches, UM cells weakly expressing CD-146 might not be enriched, leading to inaccurate results. Although CD146 tends to be more strongly expressed in metastatic UM cells [52], the mesenchymal transition of metastatic UM cells remains poorly investigated. This limitation could explain why previous studies have used other immunoaffinity- and immunomagnetic-based isolation methods to capture UM CTCs. In the literature, large differences have been found depending on the immunomagnetically embedded antibody used. Tura et al. used a dual-immunomagnetic enrichment (NKI/C3, NKI/beteb immunobeads) and identified CTCs in 93.6% of primary UM patients [57]. Ulmer et al. [58] and then Suesskind et al. [59] used the anti-melanoma-associated chondroitin sulphate proteoglycan (MCSP) antibody conjugated with microbeads and identified CTCs in 19% and 14% of UM patients, respectively. Eide et al. used sheep anti-mouse IgG antibody-coated superparamagnetic particles conjugated with several antibodies (9.2.27 anti-melanoma-associated antibody, Ep-1 IgG1 antibody, and 376.96 antibody) and found CTCs in only 1.6% (n = 328) of UM patients [60]. More recently, Beasley et al. used a multimarker panel of immunomagnetic beads directed against anti-ABCB5, anti-gp100, anti-CD146, and anti-MCSP antibodies and identified CTCs in 86% of patients with localized UM (n = 43) [54].

### 3.2. Other Features to Be Considered

Although paramount, the recovery rate (sensitivity) is insufficient to assess a CTC platform. In our opinion and experience, the ideal CTC platform should have the following characteristics: (i) reliable with satisfactory sensitivity and specificity rates, (ii) reproducible between multiple experiments, (iii) user-friendly and not requiring extensive training, (iv) capable of capturing CTCs regardless of their phenotypic characteristics, (v) able to process several samples simultaneously, (vi) allow downstream analyses to be performed (genetics, proteomics, culture, CDX), (vii) associated with rapid processing to minimize the time between blood collection and CTC isolation, (viii) cost attractive, and (ix) ergonomic. Overall, the goal of a CTC platform is to be used in daily clinical practice (Table 4).

Capturing CTCs is considered a time-consuming procedure, which limits its daily clinical application. The low throughput and the inability to process large sample volumes are two well-established limitations of physical microfluidic platforms [24]. Indeed, the Vortex (VTX-1) and ClearCell FX platforms can only process one sample at a time compared to ISET (up to four samples simultaneously) and Cellsearch (up to eight samples simultaneously). Of the four CTC platforms compared, ISET was the fastest device, achieving filtration in just 1–2 min. However, the prefiltration time, including thawing of RareCells Buffer, PHmetry [61], and the time for the manual transfer of pores to a plate for immunocytochemistry, lengthened the process. Based on our experience, the processing time was similar for all the CTC platforms and other CTC detection methods.

Another feature to take into account is the time required to process the samples. A significant difference between the processing times may induce a confounding bias. In our study, all samples were processed within 2 h. In clinical studies, the time to process a sample mainly depends on the collector tube used to collect the blood rather than on the platform itself. Usually, CTCs are processed within a few hours and up to 72 h, depending on the blood collection tubes used. CTCs isolated with EDTA tubes should be processed within 24 h, whereas streck or Cellsave tubes allow the analysis to be delayed for up to 72 h [33,62]. This feature is of prime interest for future multicentric studies in which a longer processing time will be required.

Good reproducibility is essential for application in daily clinical practice [24]. In our experience, automation is paramount and manual processing should be reduced as much as possible to reduce the interoperator variability. All the platforms we compared were relatively user-friendly. In our experience, CTC identification with the ISET platform, which consisted of manually cutting the pores, was a bit tedious and time-consuming. Due to manual errors, we had to repeat the experiments several times to achieve n = 13. In our experience, the Vortex (VTX-1) device combined several qualities such as a simple dilution with sterile PBS, automatic filtration, and collection of CTCs in a PBS solution, allowing simple and rapid plate fixation before automated immunocytochemistry.

Recently, the counting of CTCs has been supplanted by their molecular characterization [24]. Downstream analyses using the Vortex (VTX-1) [63,64,65] and ClearCell FX [66] platforms have been published. Recovering CTCs using Cellsearch is often considered difficult due to the strong immunomagnetic interactions with the platform surface [22]. Despite this, several articles have demonstrated the feasibility of genetic analyses and even of a CTC-derived explant using the Cellsearch platform [67]. Molecular analyses have also been performed using the ISET technology [68,69,70] as well as the collection of live cells using the Rarecells protocol [61]. In our experience, downstream analyses are easier to perform when CTCs are freely collected in liquid as is the case with the Vortex (VTX-1) and ClearCell FX devices.

Although it has not been investigated in detail due to a very low recovery rate, the Cellsearch device is considered the most specific CTC device, yielding the best purity. This can be explained by the use of positive immunomagnetic enrichments [24]. In addition, specific training is mandatory and provided by the manufacturer. This probably limits the interoperator variability and the false-positive detection rates [71]. In contrast, size-dependent isolation techniques, as with the ISET platform, are associated with very low purity rates [24].

Collecting viable CTCs has become a priority in order to offer personalized medicine. Patient-derived xenografts (PDX) from previously isolated CTCs have been reported with the Vortex (VTX-1) [72], ISET [61] and Cellsearch [67] platforms. The chemosensitivity of previously isolated CTCs has been reported in vitro with the ClearCell FX device [73]. Inertial focusing technologies (e.g., with the Vortex (VTX-1) and ClearCell FX) are known to exert little mechanical stress on CTCs and are thought to yield a higher rate of viable CTCs compared to other technologies [24].

Today, laboratory ergonomics has become an issue and even a priority [74], particularly in our department, which offers additional biobanking activities [75]. The four CTC capture devices are ergonomic. CTC identification was performed using the same immunohistochemistry platform (Benchmark Ultra, Roche Diagnostics, Tuczon, AZ, USA) for the Vortex (VTX-1), ClearCell FX, and ISET devices. This immunohistochemistry platform is commonly used in daily clinical practice and is present in many pathology laboratories, and therefore requires no additional space in our workplace. The Cellsearch device comprises a CTC tracker to capture CTCs (Automated Celltracks Autoprep system) and a large CTC analyzer dedicated to CTC revelation (Celltracks Analyzer II system), which is not used in our daily clinical practice.

The price of the device, dedicated equipment, and after-sales service will not be discussed because our devices were acquired at different dates and contracts are generally negotiated on a case-by-case basis. A careful health economics analysis investigating the impact of LB in UM with the aim of providing dedicated and personalized medicine would be of strong interest in the near future [76].

As already mentioned, the choice of the ideal CTC platform differs greatly from one team to another [24]. For diagnostic/prognostic purposes (CTC counting), a high recovery rate (sensitivity) as well as rapid processing are mandatory, as offered by physical and label-free platforms (e.g., the ISET), or by the FDA-approved Cellsearch in metastatic breast, colorectal, and prostate cancers [18]. For drug testing and CDX experiments, viable CTCs are needed and label-free platforms based on the inertial focusing process should be preferred (e.g., the Vortex (VTX-1) and ClearCell FX).

### 3.3. Strengths of the Study and Perspectives

Our study was the first to compare the efficiency of four different commercially available CTC platforms using a UM cell line. Using a single UM cell line allowed a direct comparison to be made and the experimental conditions to be standardized, thus limiting variability due to the cells and experimental conditions. Only one study has compared ctDNA levels with the CTC count using the Cellsearch platform [73]. In other solid malignancies, only a few studies have directly compared various CTC isolation techniques [29].

We developed a standardized and reproducible methodology for collecting and detecting CTCs using the Vortex (VTX-1) and ClearCell FX microfluidic platforms. Melan-A red immunocytochemistry was performed using the automated BenchMark Ultra system (Roche Diagnostics, Tuczon, AZ, USA), which is used in daily clinical practice. In our opinion, this allows a reduction in the interoperator variability.

It is estimated that a tumor lesion detectable using conventional imaging techniques is composed of about 10^9^ cells [48,77]. Therefore, CTC identification prior to the diagnosis of metastatic spread could allow early diagnosis and “preventive” treatment [17]. This is of prime interest in UM since no curative treatment is currently available for metastatic UM [10]. Therefore, treatment administered at a very early stage, when CTCs are circulating, could represent a promising therapeutic approach in the near future.

The number of CTCs recovered has been associated with a poorer prognosis in UM. Anand et al. observed a greater number of CTCs in patients with metastatic disease and demonstrated that the number of CTCs correlated with class 2 UM [49]. Mazzini et al. found that the detection of >10 CTCs was significantly associated with aggressive histological types and reduced overall survival [39]. Bidard et al. found that the number of CTCs correlated with the presence of miliary liver metastases [48]. The loss of chromosome 3 and the loss of BAP1 nuclear expression have been strongly associated with the metastatic spread of UM [78,79]. Interestingly, Tura et al. studied chromosome 3 status in CTCs isolated from 44 non-metastatic UM patients using an immuno-FISH isolation technique (anti-NKIC3 and anti-MCSP antibodies) [57]. They found that CTCs with chromosome 3 monosomy were significantly associated with a higher TNM stage than CTCs with chromosome 3 disomy. An interesting perspective would be to monitor BAP1 expression along with Melan-A expression by multiplex immunocytochemistry in order to identify CTCs and obtain additional prognostic insights. The prognostic power of CTC identification and the loss of BAP1 expression could help to predict with high accuracy the metastatic risk in a patient with primary UM.

### 3.4. Limitations

This study has several limitations. First, we compared four different devices at different time periods (study period was from January 2021 to December 2022) and over several weeks (except for Cellsearch, where all the experiments were carried out the same day). However, it was impossible to realize all the analyses on the same day because the Vortex VTX1 and ClearCell FX instruments require sample processing one by one. In addition, it took a while to order the different buffers, restart some platforms, and unfreeze the cells before experiments with a new platform. Similarly, we compared four platforms with different throughputs and therefore different processing times. CTC filtration with the ISET device was very fast (≈1–2 min) compared to the other three CTC platforms (≈60 min). It could be assumed that the longer the processing time, the lower the cell viability and the lower the recovery rate. We tried to reduce this potential bias by using a specific control (see Section 4) in each experiment to determine an adjusted recovery rate (Table 1). We did not observe any significant differences between the non-adjusted and adjusted recovery rates. We even observed an improvement in the adjusted recovery rate with the ISET device, suggesting that this bias was very limited. All the experiments were carried out by the same operator and we used automated devices as much as possible to increase the daily clinical application of our protocol. Therefore, we do not think that intravariability represents a major issue in this study.

Secondly, our study was an in vitro study, which could be considered a limitation or an advantage. On the one hand, CTC viability is thought to be higher in venous blood due to continuous interactions with the microenvironment (other blood cells such as neutrophils and platelets, cytokines, growth factors, etc.) [80] compared to in vitro conditions in RPMI media where the cells are more likely to die from anoikis [22,80]. The in vitro handling of tumor cells in culture could alter their viability. This could have led to an underestimation of the actual recovery rate in our in vitro assessment. We tried to take this into account by performing several experiments using venous blood. In addition, in vivo UM CTCs may carry several mutations (frequent chromosome 3 monosomy, loss of BAP1) and surface marker changes during the EMT [64], which probably distinguishes them from other commercially available UM cell lines. On the other hand, only in vitro experiments allow the recovery rate of a given CTC platform to be assessed with a limited risk of false positives. Indeed, when processing a patient’s blood, it is impossible to accurately assess the initial number of venous CTCs, and several cells may be wrongly identified as CTCs (false positives). Therefore, it is essential to distinguish between the recovery rate, which can only be assessed in in vitro experiments, and the detection rate, which is assessed in in vivo experiments conducted in patients. Most studies have found variable CTC detection rates using different CTC platforms (Table 3) [24]. For reproducibility purposes, we chose to systematically target Melan-A in all our experiments, except for the Cellsearch device (automatic detection of the HMW-MAA and CD146). Melan-A is expressed in primary and metastatic UM, thus limiting an underestimation bias [81].

Thirdly, we only assessed the sensitivity of each platform by determining a specific recovery rate. However, the specificity or purity of the devices was NOT calculated. Indeed, the number of CTCs collected/total number of cells collected was not assessed [24]. Purity has been shown to be essential for downstream analyses, such as genomics and proteomics, by reducing confounding DNA or RNA from leukocytes [24].

Fourthly, the experiments performed with the Cellsearch device were inconclusive. The recovery rate was low, as discussed above. Our results differ from those previously published for Cellsearch in UM patients. The most plausible explanation would be the low expression of CD146 in our OMM 2.3 cells. This highlights a limitation of the Cellsearch device and of all label-dependent capture devices.

Fifthly, the initial number of cells seeded was not strictly identical between the experiments, the lowest number being observed when assessing the ISET and Cellsearch devices (Table 2). However, as the ISET was associated with the best recovery rates, we do not believe that this feature was of primary interest. In addition, we seeded about 15–150 cells per experiment, which is lower than previously reported (≈300) in other in vitro studies [37].

Finally, only positive controls but no negatives controls were performed in this study. Negative controls are not useful for experiments conducted in RPMI but would have been relevant for experiments conducted in venous blood.

## 4. Materials and Methods

### 4.1. Study Protocol

An in vitro study was conducted at Nice University Hospital between January 2021 and December 2022. For each CTC platform, 10–13 experiments were performed depending on the CTC device used after seeding a definite number of metastatic UM cells (OMM 2.3 cell line) diluted in RPMI medium and/or venous blood from a healthy donor. The efficiency of each CTC platform was determined by calculating the cell recovery rate. A dedicated control was used to obtain an adjusted recovery rate when possible. The whole protocol is summarized in Figure 3.

#### 4.1.1. Cell Preparation

The metastatic OMM 2.3 (also called OMM 2.5) UM cell line was used in all the experiments. OMM 2.3 cells were kindly provided by Dr. Jager MJ from Leiden University Medical Center (Leiden, The Netherlands). OMM 2.3 cell characteristics are summarized in Table 5 [82,83]. Our OMM 2.3 cell line was externally authenticated (Northgene, UK, available on request).

OMM 2.3 cells were used to determine the efficiency of four CTC platforms. OMM 2.3 cells were cultured in RPMI medium supplemented with 10% fetal bovine serum (FBS) and 1% Penicillin-Streptomycin (Gibco, Thermofisher, Scientific, Waltham, MA, USA) at 37 °C in a humidified atmosphere containing 5% of CO_2_. When a confluence of 70% was reached, the OMM 2.3 cells were stained with the Celltracker green CMFDA dye (Thermofisher Scientific, Waltham, MA, USA) at a concentration of 1 µM (Figure 4A). After 30 min, the medium and the dye were removed by aspiration. TrypLE Express (Thermofisher Scientific, Waltham, MA, USA) was used to dissociate the cells. As shown in Figure 4B, the mean diameter of OMM 2.3 cells was ≈20–25 µm (the diameter of a CTC and a white blood cell is 12–25 µm and 8–14 µm, respectively [24]).

The cells were successively diluted in RPMI medium supplemented with FBS. The cells were seeded in a 96-well plate (≈100–150 µL/well). Cells were manually counted twice by the same operator using an inverted EVOS fluorescent microscope. Wells containing ≈15–150 cells were used immediately for further experiments with the CTC platforms. The cells were added to 1.5 mL Eppendorf tubes (Figure 5). Empty wells were counted again to ensure that no residual cells adhered to the plate. If any cells had not detached from the plate, the latter were removed from the cell count. The *initial cell number* (iCN) was calculated as follows: cell count in Eppendorf tube minus cells stood attached to the plate. In all experiments and all the protocol steps, all the cones and tubes were coated with RPMI medium supplemented with FBS to avoid cell adherence to their walls. OMM 2.3 control cells were counted and then added to 1.5 mL Eppendorf tubes containing RPMI, as described above. Control cells were processed simultaneously. For experiments performed in venous blood, a venous puncture was performed on the same day and blood was collected in a 10 mL EDTA tube for the Vortex (VTX-1), ClearCell FX, and ISET devices or in a 10 mL Cellsave tube for the Cellsearch device (Figure 5). The total duration of cell preparation was about 90 min.

#### 4.1.2. OMM 2.3 Cell Capture

In this study, 4 different CTC platforms were compared. For the Vortex (VTX-1), ClearCell FX, and ISET platforms, 10 and 3 experiments were performed on freshly isolated OMM 2.3 cell suspensions in RPMI medium and venous blood from a healthy donor, respectively. The Cellsearch experiments can only be performed on blood samples, so 10 experiments were performed in venous blood.

The Vortex (VTX-1) device (Biosciences, Pleasanton, CA, USA) is a microfluidic platform based on the inertial focusing process. Cells were diluted into 40 mL of sterile PBS. When healthy blood was used, 4 mL of blood was diluted into 36 mL of sterile PBS as recommended by the manufacturer [37]. Two cycles were performed as recommended by the manufacturer for a total duration of ≈50 min [37]. Performing 2 cycles has been shown to be associated with a good balance between the purity and the recovery rate [37]. About 300 µL of filtered OMM 2.3 cell suspension was collected in PBS in a dedicated 1.5 mL Eppendorf tube. An FBS + RPMI-coated cone was used to incubate the cells with the PreservCyt solution (ThinPrep, Hologic, Villepinte, France; 20 mL) for fixation. The cells were immediately transferred to a 96-well plate using the Thinprep 2000 platform (Hologic, France).

ClearCell FX (ClearBridge, Biolidics, Singapore) is another microfluidic platform based on the inertial focusing process [42]. Cells were resuspended in ClearCell FX resuspension buffer. For blood experiments, a red cell lysis buffer (G-Bioscience, St Louis, MO, USA) was added to the 7.5 mL of blood at a 4:1 ratio (vol/vol), followed by centrifugation (1500 rpm for 10 min at room temperature) as recommended by the manufacturer [41]. The supernatant was discarded, and 4 mL of ClearCell FX resuspension buffer was added. In all experiments, a single run was performed for a total duration of ≈45 min, allowing approximately 12 mL of filtered cells to be collected in PBS. The cells were transferred, using coated cones, to the PreservCyt solution (20 mL) and processed as described above.

The ISET platform (Rarecells, Paris, France) captures cells according to their size. OMM 2.3 cells were incubated in 10 mL of venous blood from healthy donors. The blood was then diluted with the Rarecells buffer solution with a 10-fold dilution as previously described [61]. The Rarecells cartridge was then used to filter the diluted blood through the dedicated filter containing 10 pores of 8 µm size each with a 10 kPa pressure. Sterile PBS was used before and after filtration as recommended by the manufacturer [61]. Filtration did not exceed 1–2 min per sample. The 10 pores were manually fixed on a Superfrost Slide (Thermofisher Scientific, Waltham, MA, USA) before performing immunochemistry.

Cellsearch (Menarini Silicon Biosystems, Florence, Italy) is the only FDA-approved CTC device, and was approved in 2004. Cellsearch’s technology is based on immunomagnetic capture with positive and negative enrichments [18]. OMM 2.3 cells were incubated in 7.5 mL Cellsave tubes with venous blood from healthy donors. No in vitro analyses with RPMI could be performed with the Cellsearch device because a blood interface is needed to launch the device. Therefore, 10 experiments were performed using venous blood from healthy donors. A control (M229 cell line) derived from cutaneous melanoma was used, and the expression of CD146 was also assessed to ensure the proper functioning of the Cellsearch device. The manufacturer externally processed the Cellsave tubes within 72 h. Briefly, CD146^+^ cells were immunomagnetically enriched and stained with 4,2-diamidino-2-phenylindole dihydrochloride (DAPI), a phycoerythrin-conjugated antibody (clone 9.2.27) that binds to HMW-MAA, and a mixture of two allophycocyanin-conjugated antibodies to identify leukocytes (CD45, clone HI30) and endothelial cells (CD34, clone 581) [48,51]. Oval-shaped cells positive for DAPI and HMW-MAA and negative for CD45 and CD34 were considered OMM 2.3 cells.

Controls. When possible, a dedicated control was used with the Vortex (VTX-1), ClearCell FX, and ISET devices to confirm cell viability and better assess each platform’s recovery rate. OMM 2.3 cells were cultured in ≈300–400 µL of RPMI medium supplemented with FBS). OMM 2.3 control cells were fixed in the PreservCyt solution (20 mL) using FBS-coated cones at the same time as the processing of the tested samples. This “simultaneous” fixation allowed a reduction in any bias related to cell viability and cell lysis due to the duration of the experiment. The control cells were then seeded into a 96-well plate using the Thinprep platform (Hologic, France), as described above.

#### 4.1.3. Immunocytochemistry

Immunocytochemistry was performed using the BenchMark Ultra automated staining instrument (Roche Ventana Medical Systems, Tucson, AZ, USA) for cells recovered with the Vortex (VTX-1), ClearCell FX, and ISET platforms. An anti-melan-A (A103 clone, ready-to-use, Roche Ventana, Tucson, AZ, USA) antibody was used. The Fast Red IHC Detection Kit (Roche Ventana, Tucson, AZ, USA) was used according to the manufacturer’s recommendations for cells recovered using the Vortex (VTX-1) and ClearCell FX platforms. Sections were counter-stained with hematoxylin and bluing reagent. For cells recovered with the ISET platform, an anti-melan-A antibody and the Ultraview Universal DAB Detection Kit (Ventana, Tucson, AZ, USA) without Fast Red IHC Detection Kit were used. Each IHC staining included positive and negative controls.

#### 4.1.4. Cell Counting and Recovery Rate

The final cell number (fCN) was determined manually by 2 blinded investigators using a dedicated microscope (AM, SL). The fCN was the mean of the 2 counts. A cell was considered a captured OMM 2.3 UM cell if it had (i) a round-shaped appearance, (ii) a high nucleus/cytoplasmic ratio, and (iii) was positive for Melan-A or Melan-A red staining (Figure 2). The recovery rate (RR) was calculated as the final cell number (fCN) divided by the initial cell number (iCN). The adjusted recovery rate (aRR) was calculated as the RR obtained with the CTC platforms divided by the RR of the corresponding controls. For the Cellsearch device, the CTC count was automatically determined by the manufacturer, so no aRR could be calculated.

### 4.2. Statistics

Descriptive statistics are presented as numbers and percentages. Data normality and heteroskedasticity were assessed using the Shapiro–Wilk test and Levene’s test. The difference between the initial and final cell counts, as well as the recovery rates depending on the CTC platform used, were assessed using the Kruskal–Wallis test. If the null hypothesis of the Kruskal–Wallis test was rejected, post hoc pairwise analyses were performed using the Dunn–Bonferoni test. The alpha risk was set at 5% (α = 0.05). Statistical analyses were performed with EasyMedStat (version 3.22; www.easymedstat.com, accessed on 12 April 2023).

## 5. Conclusions

We found the highest tumor cell recovery rates with the ISET and Vortex (VTX-1) platforms compared to the ClearCell FX and Cellsearch devices using our UM cell line. However, other parameters such as the reason for choosing a platform (diagnosis, genetics, drug sensitivity, CDX models), reproducibility, purity, user-friendliness, cost-effectiveness, and ergonomics should also be taken into account before using these devices in daily clinical practice.

## Figures and Tables

**Figure 1 ijms-24-11075-f001:**
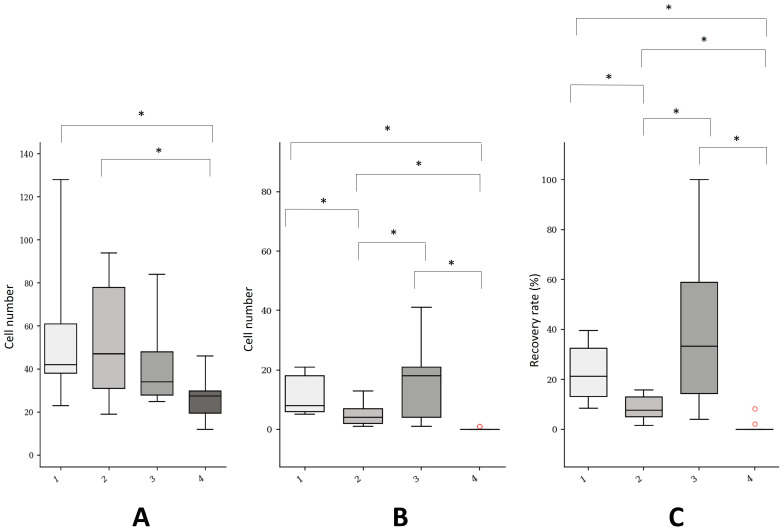
Summary of the CTC recovery rates of four CTC isolation platforms using the OMM 2.3 UM cell line. (**A**) initial cell number (iCN); (**B**) final cell number (fCN); (**C**) overall recovery rate (in RPMI + venous blood); 1: Vortex (VTX-1); 2: ClearCell FX; 3: ISET; 4: Cellsearch. (*) statistically significant.

**Figure 2 ijms-24-11075-f002:**
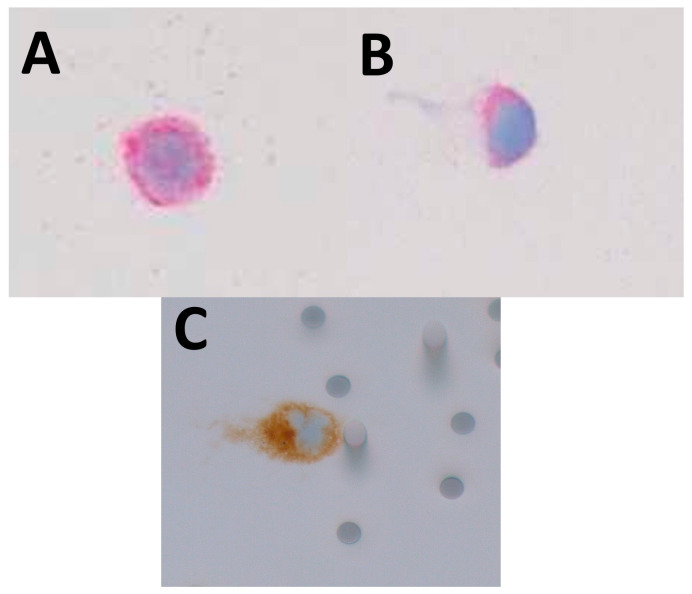
Illustrative photographs of the CTC collected using the (**A**) Vortex (VTX-1); (**B**) ClearCell FX; (**C**) and ISET devices. OMM 2.3 cells were stained with Melan-A red (**A**,**B**), and with Melan-A (**C**). ((**A**–**C**): immunohistochemistry; Melan-A; A103 clone, ready to use, Roche Ventana, Tucson, AZ, USA; original magnification ×400).

**Figure 3 ijms-24-11075-f003:**
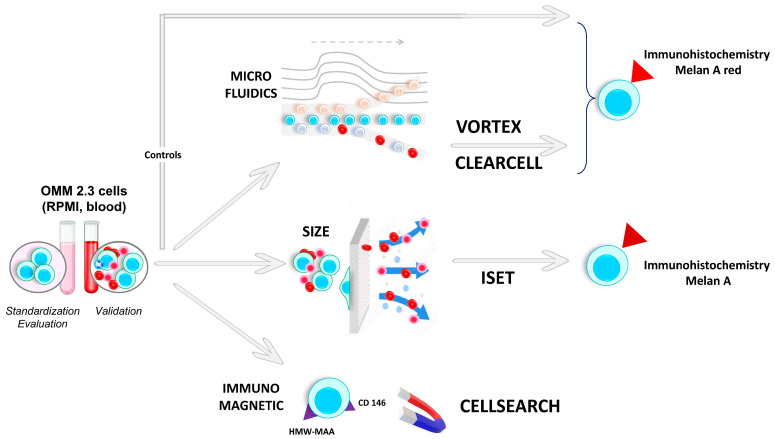
Summary of the whole protocol. HMW-MAA: high-molecular-weight melanoma-associated antigen.

**Figure 4 ijms-24-11075-f004:**
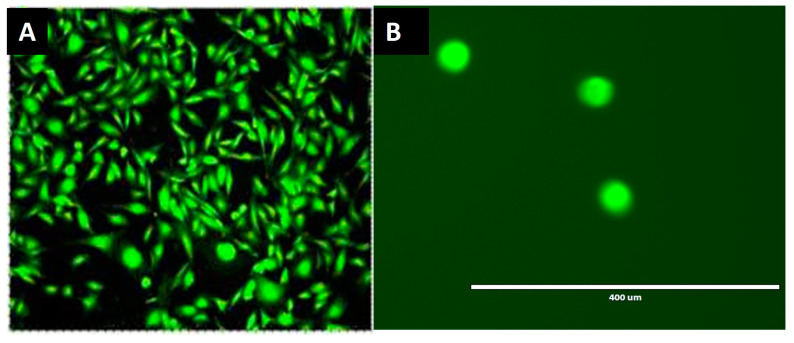
(**A**) Staining of OMM 2.3 cells with the Celltracker green CMFDA dye (magnification ×60). (**B**) The size of OMM 2.3 cells after dissociation with TrypLE Express was approximately 25 µm.

**Figure 5 ijms-24-11075-f005:**
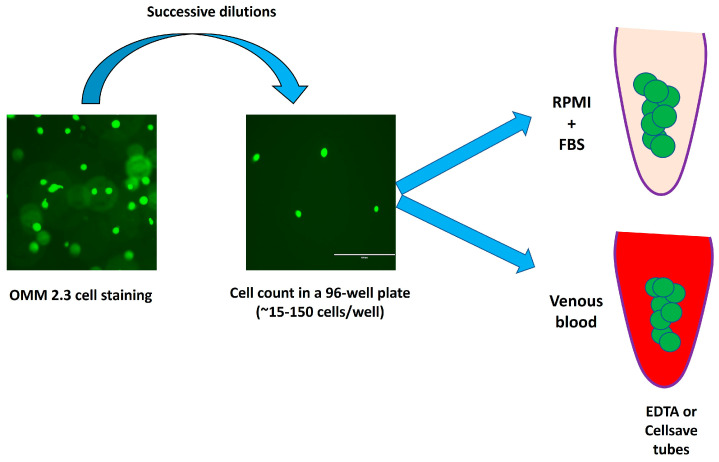
OMM 2.3 UM cell preparation (magnification ×60). FBS: fetal bovine serum.

**Table 1 ijms-24-11075-t001:** Summary of the various analytical phases of CTC screening [22,24].

Phase	CTC Capture	CTC Identification	CTC Downstream Analyses
Examples	Immunoaffinity with positive or negative enrichment (immunomagnetic, microfluidic) Biophysical (“label-free”): Size Deformability Density Microfluidics Electrophoresis	Immunohistochemistry Immunofluorescence Flow cytometry Spectrophotometry Electrical impedance	Genomics Transcriptomics Proteomics CTC culture Cell line-derived xenografts (CDX)

**Table 2 ijms-24-11075-t002:** CTC recovery rates obtained with four CTC isolation platforms. iCN: initial Cell Number; fCN: final Cell Number; NA: Non-Applicable.

CTC Platform	Mean iCN ± SD [Range]	Mean fCN ± SD [Range]	Overall Recovery Rate	Recovery Rate	Adjusted Recovery Rate	Recovery Rate	Adjusted Recovery Rate
			in RPMI + Venous Blood	in RPMI Alone (n = 10)		in Venous Blood Alone (n = 3)	
Vortex (VTX-1) (n = 13)	57.31 ± 33.93 [23.0; 128.0]	11.31 ± 6.14 [5.0; 21.0]	22.2%	22.6%	25.6%	20.7%	26.6%
ClearCell FX (n = 13)	54.46 ± 26.43 [19.0; 94.0]	4.85 ± 3.93 [1.0; 13.0]	8.9%	10.2%	15.1%	4.7%	7.2%
ISET (n = 13)	42.08 ± 19.5 [25.0; 84.0]	19.92 ± 22.39 [1.0; 84.0]	39.2%	30.3%	43.5%	69.1%	89%
Cellsearch (n = 10)	26.9 ± 11.18 [12.0; 46.0]	0.2 ± 0.422 [0.0; 1.0]	1.1%	NA	NA	1%	NA

**Table 3 ijms-24-11075-t003:** Summary of the previous studies comparing the CTC platforms used in the current study.

Authors, Year	In Vitro or In Vivo Study	Cancer Type	CTC Platform Used (Detection or Recovery Rate)
Present study	In vitro	Uveal melanoma	Vortex (VTX-1) (22.2%) ClearCell FX (8.9%) ISET (42.2%) Cellsearch (1.1%)
Tamminga et al. 2020 [26]	In vivo	Non-small cell lung carcinoma	ISET (88%) Cellsearch (69%)
Yin et al. 2019 [27]	In vivo	Breast cancer	ClearCell FX (27.8%) Cellsearch (27.8%)
Bai et al. 2018 [28]	In vivo	Renal cell carcinoma	ISET (36.1%) Cellsearch (19.4%)
Che et al. 2016 [37]	In vivo	Breast and lung cancer	Vortex (VTX-1) (85%) Cellsearch (15%)
Kallergi et al. 2016 [29]	In vitro	Breast cancer	ISET (95%) Cellsearch (52%)
Li et al. 2015 [38]	In vivo	Esophageal cancer	ISET (32.8%) Cellsearch (1.8%)
Morris et al. 2014 [30]	In vivo	Hepatocellular cancer	ISET (100%) Cellsearch (28%)
Krebs et al. 2012 [31]	In vivo	Non-small cell lung carcinoma	ISET (80%) Cellsearch (23%)
Khoja et al. 2012 [32]	In vivo	Pancreatic cancer	ISET (93%) Cellsearch (40%)
Hofman et al. 2011 [33]	In vivo	Non-small cell lung carcinoma	ISET (50%) Cellsearch (39%)

**Table 4 ijms-24-11075-t004:** Comparison of the four CTC platforms used in the present study.

CTC Platform	Mechanism	Reliability (Capture Rate) in the Present Study	Reproducible and User-Friendly	Total Duration in Minutes (CTC Capture and Identification)	Number of Samples That Can Be Processed at the Same Time	Ease of Performing Downstream Analyses	Ergonomic
Vortex (VTX-1)	Microfluidic	22.2%	Yes	~180	1	Easy	Yes
ClearCell FX	Microfluidic	8.9%	Yes	~180	1	Easy	Yes
ISET	Isolation according to the size	39.2%	Variable	~150	4	More difficult	Yes
Cellsearch	Immunomagnetic	1.1%	Yes	~180	8	More difficult	No

**Table 5 ijms-24-11075-t005:** Summary of the OMM 2.3 cell line characteristics.

OMM 2.3 Cell Characteristics	
Age of the donor	80 years old
Gender of the donor	Male
Primary treatment	Iodine-125 plaque followed by secondary enucleation
Cellularity	Mixed with a predominance of epithelioid cells (70%)
Ciliary body involvement	Yes
Mitotic index	11 mitotic figures per ten 40× high-power fields
Necrosis	Yes (post-radiation plaque)
Extraocular extent	Yes, through the emissary venous channels
Time to metastases	8 years
Time between liver biopsy and death	1 month
Chromosome 3 status	Disomy 3
BAP 1 status	Expressed
Melan-A expression	80% of cells

## Data Availability

Not applicable.

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
