# Peer review of "Assessment of Different Circulating Tumor Cell Platforms for Uveal Melanoma: Potential Impact for Future Routine Clinical Practice"

_ijms, 2023, doi:10.3390/ijms241311075_

Round 1
Reviewer 1 Report
This manuscript compared the efficiency of 4 commercially available CTC platforms (Vortex (VTX-1), ClearCell FX, ISET, and Cellsearch) for the detection and identification of uveal melanoma cells. The overall recovery rates and other features such as the reason for choosing a platform, reproducibility, purity, etc. were also compared and discussed in detail. Overall, this work showed great practical significance, especially in clinical applications, and may provide a broad reading interest to the journal readers. Therefore, I think this manuscript can be accepted for publication in International Journal of Molecular Sciences.
Author Response
On behalf of all the authors, we would like to thank the reviewer for the time spent for reviewing our work.
Reviewer 2 Report
The presented manuscript aims to compare the performance of different systems to isolate CTC for uveal melanoma patients. For comparison, authors tested a uveal melanoma cell line in healthy donors' blood and RPMI. The main criticism of the study is the different amount of starting cells concerning the different devices and the use of a unique UM cell line for CTC detection comparison. Anyway, the manuscript is well-written and organized.
Hereafter are my concerns:
1. It is not clear if the experiments were carried out in once analyzing different vials/tubes (excluding CellSearch because of outsourcing) or repeated on different days. If the latter, please provide the intertest variability. Also, the intra-test variability would be appreciated for each device.
2. Figure 3 and Table 1 report CTC downstream analysis as post analytical phase. I disagree; they are analytical phases. Please correct accordingly.
3. It is unclear to me how the ten blood samples were organized. How many samples of blood have been used? 10 tubes of 10 ml blood for each platform? Did the blood come from the same donor? Or multiple donors?
4. I also appreciate a careful explanation of the experiments in the RPMI medium.
5. Three CEN technical specifications were published for CTC pre-analytical processes in 2021, referring to RNA isolation (CEN TS 17390-1), DNA isolation (CEN TS 17390-2) and staining (CEN TS 17390-3). Which of them has been followed? If none, please explain why and comment on the use of technical specifications for pre-analytical procedures for CTC analysis.
6. Sensitivity regarding the minimum number of detectable cells per device would also be helpful. (At least excluding the CellSearch)
7. The abstract should include, in addition to the % of recovery also, the pure number of recovered cells for each platform.
8. Figure 1: y-axis unit of measure was not included.
9. Although different uveal cell lines are available, why OMM 2.3 cells were chosen? Please, describe the rationale for selecting them.
10. Although underestimation is the primary concern in CTC analyses, did the authors also run negative controls?
11. The supplementary figure is in the wrong place.
English is fine, only some minor typing errors should be checked.
Author Response
On behalf of all the authors, we thank the reviewer for his/her valuable work and remarks. Please find below our comments and answers for each point. Corrections have been highlighted in yellow in the revised manuscript.
The presented manuscript aims to compare the performance of different systems to isolate CTC for uveal melanoma patients. For comparison, authors tested a uveal melanoma cell line in healthy donors' blood and RPMI. The main criticism of the study is the different amount of starting cells concerning the different devices and the use of a unique UM cell line for CTC detection comparison. Anyway, the manuscript is well-written and organized.
Hereafter are my concerns:
- It is not clear if the experiments were carried out in once analyzing different vials/tubes (excluding CellSearch because of outsourcing) or repeated on different days. If the latter, please provide the intertest variability. Also, the intra-test variability would be appreciated for each device.
Thank you for your comment and important suggestions. Only the CellSearch analyses were carried out the same day. For the experiments made with the three other CTC devices, they were conducted over a 1 to 4 months period between January 2021 and December 2022. It was not possible to perform all the analyzes on the same day because the Vortex VTX1 and ClearCell FX instruments require sample processing one by one. Moreover, in daily practice, it took us times to order buffers, restart some platforms and unfreeze the cells either. This aspect has been highlighted now in the revised manuscript.
Moreover, each platform had different processing durations. This bias was discussed in the discussion (first point). To address this possible inter-test variability bias, we used a dedicated control for each processed sample. These controls allowed us to ensure the UM cell viability. The adjusted recovery rate (please see Table 2) considered the controls and results were by far similar to the non-adjusted recovery rates.
All the experiments were carried out each time by the same operator. It was important in this study to use automated devices (see Material section) to increase the potential daily clinical practice of these approaches. Therefore, we do not think that intra-test variability is a significant bias in our study. This was now added in the discussion of the revised manuscript.
- Figure 3 and Table 1 report CTC downstream analysis as post analytical phase. I disagree; they are analytical phases. Please correct accordingly.
Thanks for this important comment. To avoid any confusion, the term post analytical phase has been removed accordingly from Figure 3, Table 1 and from the manuscript.
- It is unclear to me how the ten blood samples were organized. How many samples of blood have been used? 10 tubes of 10 ml blood for each platform? Did the blood come from the same donor? Or multiple donors?
Three blood samples were taken from the same healthy donor for each of the three Vortex, Clearcell and Iset platforms. 4ml, 7.5ml and 10ml of venous blood were used to dilute the UM cell line as recommended by the manufacturers of Vortex, Clearcell and Iset respectively. For the CellSearch platform, only experiments on venous blood were carried out. No RPMI experiments could be performed with the CellSearch platform because a blood interface is absolutely necessary to start the CellSearch. This is developed now in detail in the Material and Method section.
- I also appreciate a careful explanation of the experiments in the RPMI medium.
Thank you for this commentary. All the experiments are now detailed in the Material method and in Figure 5.
Briefly
- BEFORE CTC PLATFORM
The cells were successively diluted in RPMI medium supplemented with FBS. The cells were seeded in a 96-well plate (≈100-150 µL/well). Cells were manually counted twice by the same operator using an inverted EVOS fluorescent microscope. Wells containing ≈15-150 cells were used immediately for further experiments with the CTC platforms. The cells were added to 1.5-mL Eppendorf tubes (Figure 5).
- CTC PLATFORM
For the Vortex: Cells were diluted into 40 mL of sterile PBS. Two cycles were performed as recommended by the manufacturer for a total duration of ≈50 min. About 300 µL of filtered OMM 2.3 cell suspension were collected in PBS in a dedicated 1.5-mL Eppendorf tube. A FBS+RPMI-coated cone was used to incubate the cells with the PreservCyt solution (ThinPrep, Hologic, France; 20 mL) for fixation. The cells were immediately transferred to a 96-well plate using the Thinprep 2000 platform (Hologic, France).
For the ClearCell FX: Cells were resuspended in ClearCell FX resuspension buffer. In all experiments, a single run was performed for a total duration of ≈45 minutes, allowing collecting approximately 12 mL of filtered cells in PBS. The cells were transferred, using coated cones, to the PreservCyt solution (20 mL) and processed as described above.
For the ISET platform: The Rarecells cartridge was then used to filter the cells through the dedicated filter containing 10 pores of 8-µm size each with a 10 kPa pressure. Sterile PBS was used before and after filtration as recommended by the manufacturer The 10 pores were manually fixed on a Superfrost Slide before performing immunochemistry.
- Three CEN technical specifications were published for CTC pre-analytical processes in 2021, referring to RNA isolation (CEN TS 17390-1), DNA isolation (CEN TS 17390-2) and staining (CEN TS 17390-3). Which of them has been followed? If none, please explain why and comment on the use of technical specifications for pre-analytical procedures for CTC analysis.
Thank you very much for this comment. Indeed, we followed the recommendations of each supplier for the pre-analytical phase. We then decided to develop an ICC staining technique in our pathology department, with a view to possible subsequent application in clinical practice. To do this, we first developed the labelling of the cell line with melan A antibody. We did several tests involving a large number of cells (validation of positive cytoplasmic labelling, verification of the absence of false positivity results in the absence of secondary antibody). Once the labelling was optimal, we applied it to our experiments. For each labelling of the cells, we added a positive and a negative control per test. The technical processes carried out at the LPCE laboratory have been subject to accreditation in accordance with the ISO 15189 norm, since the LPCE laboratory is accredited for immunohistochemistry, in particular for Melan A IHC in melanoma.
- Sensitivity regarding the minimum number of detectable cells per device would also be helpful. (At least excluding the CellSearch)
Thank you for this important suggestion However, that was not the initial purpose of this work. At this stage, we demonstrated that the Vortex, Clearcell and ISET devices were able to detect at least one OMM 2.3 cell among 23, 19 and 25 seeded cells. Further analyzes devoted to minimum detectable cells would be now of strong interest in a complementary article.
- The abstract should include, in addition to the % of recovery also, the pure number of recovered cells for each platform.
The pure number of recovered cells for the 4 platforms has been added in the abstract accordingly.
- Figure 1: y-axis unit of measure was not included.
Thank you for your comment. The y-axis has been added in the revised Figure accordingly.
- Although different uveal cell lines are available, why OMM 2.3 cells were chosen? Please, describe the rationale for selecting them.
OMM 2.3 cell lines were used because they were isolated from a metastatic melanoma patient, unlike Mel 270 cell lines. For a CTC study, we considered the use of metastatic cells to be more closely related to daily clinical practice. The general characteristics of OMM 2.3 cubicles are summarized in Table 4.
|
OMM 2.3 cell characteristics |
|
|
Age of the donor Gender of the donor Primary treatment Cellularity Ciliary body involvement Mitotic index Necrosis Extraocular extent Time to metastases Time between liver biopsy and death Chromosome 3 status BAP 1 status Melan-A expression |
80 years old Male Iodine-125 plaque followed by secondary enucleation Mixed with a predominance of epithelioid cells (70%) Yes 11 mitotic figures per ten 40× high-power fields Yes (post-radiation plaque) Yes, through the emissary venous channels 8 years 1 month Disomy 3 Expressed 80% of cells |
- Although underestimation is the primary concern in CTC analyses, did the authors also run negative controls?
Negative controls have not been included. We totally agree that for experiments carried out with venous blood, negative controls would have been valuable. This limitation has been added in the discussion. We are currently conducting a clinical study with the Vortex VTX1 device and included 5 controls. None of them were positive for Melan A red.
- The supplementary figure is in the wrong place.
The supplementary figure has been removed from the manuscript.

Reviewer 3 Report
The article by Martel et al. "Assessment of different circulating tumor cell platforms for uveal melanoma: Potential impact for future routine clinical practice" covers a potentially interesting and emerging topic related to the oncological diagnostics.I regard the main point of this paper as highly attractive as well as the results are clearly presented. The text does not contain any major errors, therefore I have some minor comments and recommendations:
1. There is a need to provide slightly more expanded introduction shortly
mentioning/describingpharmacoeconomic aspects of uveal melanoma.
2. The figure summarizing and clarifying the conclusions should be added.
3. Following references should be added and properly cited within the main text:
-Mela A, Poniatowski ŁA, Drop B, Furtak-Niczyporuk M, Jaroszyński J, Wrona W, Staniszewska A, Dąbrowski J, Czajka A, Jagielska B, Wojciechowska M, Niewada M. Overview and Analysis of the Cost of Drug Programs in Poland: Public Payer Expenditures and Coverage of Cancer and Non-Neoplastic Diseases Related Drug Therapies from 2015-2018 Years. Front Pharmacol. 2020 Aug 14;11:1123. doi: 10.3389/fphar.2020.01123.
- Kanthan GL, Jayamohan J, Yip D, Conway RM. Management of metastatic carcinoma of the uveal tract: an evidence-based analysis. Clin Exp Ophthalmol. 2007 Aug;35(6):553-65. doi: 10.1111/j.1442-9071.2007.01550.x.
- Mela A, Rdzanek E, Tysarowski A, Sakowicz M, Jaroszyński J, Furtak-Niczyporuk M, Żurek G, Poniatowski ŁA, Jagielska B. The impact of changing the funding model for genetic diagnostics and improved access to personalized medicine in oncology. Expert Rev Pharmacoecon Outcomes Res. 2023 Jan;23(1):43-54. doi: 10.1080/14737167.2023.2140139.
- Konstantinidis L, Rospond-Kubiak I, Zeolite I, Heimann H, Groenewald C, Coupland SE, Damato B. Management of patients with uveal metastases at the Liverpool Ocular Oncology Centre. Br J Ophthalmol. 2014 Jan;98(1):92-8. doi: 10.1136/bjophthalmol-2013-303519.
4. In some places the use of English could be improved on.
Completing this gaps will have an impact on the understanding the aim of the study and, from my point of view, is absolutely necessary.
minor review
Author Response
On behalf of all the authors, we thank the reviewer for his/her valuable work and remarks. Please find below our comments and answers for each point. Corrections have been highlighted in yellow in the revised manuscript.
The article by Martel et al. "Assessment of different circulating tumor cell platforms for uveal melanoma: Potential impact for future routine clinical practice" covers a potentially interesting and emerging topic related to the oncological diagnostics.I regard the main point of this paper as highly attractive as well as the results are clearly presented. The text does not contain any major errors, therefore I have some minor comments and recommendations:
- There is a need to provide slightly more expanded introduction shortly mentioning/describing pharmacoeconomic aspects of uveal melanoma.
Thank you very much for this important suggestion. Therefore, we added now few sentences in the introduction section regarding the incidence as well as the latest pharmacology developments in uveal melanoma.
Modification: “UM remains rarely encountered with an annual incidence of 5 cases per million in Western countries [2]. UM is genetically different from cutaneous melanoma explaining with targeted therapies and immunotherapies are very disappointing in UM [5,7]. Recently, new hopes have been raised by the development of tebentafusp which is a TCR/Anti-CD3 bispecific fusion protein targeting gp100[8]. However, tebentafusp can only be administrated to HLA-A*02:01-positive patients, overall survival is modeste (73%versus 59%) at one year, studies with longer follow-up (at least 5 years) are warranted and a precise country by country medico-economic analysis is still lacking [9].”
- The figure summarizing and clarifying the conclusions should be added.
The conclusions are summarized in the graphical abstract which was submitted with the manuscript. We have now modified the graphical abstract to better summarize the practical conclusions of this study. The graphical abstract is presented below.
- Following references should be added and properly cited within the main text:
-Mela A, Poniatowski ŁA, Drop B, Furtak-Niczyporuk M, Jaroszyński J, Wrona W, Staniszewska A, Dąbrowski J, Czajka A, Jagielska B, Wojciechowska M, Niewada M. Overview and Analysis of the Cost of Drug Programs in Poland: Public Payer Expenditures and Coverage of Cancer and Non-Neoplastic Diseases Related Drug Therapies from 2015-2018 Years. Front Pharmacol. 2020 Aug 14;11:1123. doi: 10.3389/fphar.2020.01123.
- Kanthan GL, Jayamohan J, Yip D, Conway RM. Management of metastatic carcinoma of the uveal tract: an evidence-based analysis. Clin Exp Ophthalmol. 2007 Aug;35(6):553-65. doi: 10.1111/j.1442-9071.2007.01550.x.
- Mela A, Rdzanek E, Tysarowski A, Sakowicz M, Jaroszyński J, Furtak-Niczyporuk M, Żurek G, Poniatowski ŁA, Jagielska B. The impact of changing the funding model for genetic diagnostics and improved access to personalized medicine in oncology. Expert Rev Pharmacoecon Outcomes Res. 2023 Jan;23(1):43-54. doi: 10.1080/14737167.2023.2140139.
- Konstantinidis L, Rospond-Kubiak I, Zeolite I, Heimann H, Groenewald C, Coupland SE, Damato B. Management of patients with uveal metastases at the Liverpool Ocular Oncology Centre. Br J Ophthalmol. 2014 Jan;98(1):92-8. doi: 10.1136/bjophthalmol-2013-303519.
Thank you very much for these suggestions. We have added references 1 and 3 accordingly. We did not add the following references entitled “Management of metastatic carcinoma of the uveal tract: an evidence-based analysis” and “Management of patients with uveal metastases at the Liverpool Ocular Oncology Centre” because uveal metastases from distant carcinomas are beyond the scope of this article and may confuse non-ophthalmologist readers. Our study focused only on primary intraocular uveal melanoma. It cannot be excluded that our results would be quite different if we compared our 4 CTC platforms with carcinomas. Moreover, uveal metastases include too many malignancies to be studied in a single article. This could be of strong interest in further studies
- In some places the use of English could be improved on.
Our article has been revised by a native English editor.
Completing this gaps will have an impact on the understanding the aim of the study and, from my point of view, is absolutely necessary.

Round 2
Reviewer 2 Report
The authors responded adequately to the questions I posed.